# Pharmacokinetics and Bioequivalence of Two Empagliflozin, with Evaluation in Healthy Jordanian Subjects under Fasting and Fed Conditions

**DOI:** 10.3390/ph15020193

**Published:** 2022-02-03

**Authors:** Mohammad Hailat, Zainab Zakaraya, Israa Al-Ani, Osaid Al Meanazel, Ramadan Al-Shdefat, Md. Khalid Anwer, Mohamed J. Saadh, Wael Abu Dayyih

**Affiliations:** 1Faculty of Pharmacy, Al-Zaytoonah University of Jordan, Amman 11733, Jordan; m.hailat@zuj.edu.jo; 2Faculty of Pharmacy, Al-Ahliyya Amman University, Amman 19328, Jordan; z.zakaraya@ammanu.edu.jo (Z.Z.); ialani@ammanu.edu.jo (I.A.-A.); 3Michael Sayegh Faculty of Pharmacy, Aqaba University of Technology, Aqaba 77110, Jordan; otalmeanazel@aut.edu.jo; 4Faculty of Pharmacy, Jadara University, Irbid 21110, Jordan; rshdefat@jadara.edu.jo; 5Department of Pharmaceutics, College of Pharmacy, Prince Sattam Bin Abdulaziz University, Al-Kharj 11942, Saudi Arabia; m.anwer@psau.edu.sa; 6Faculty of Pharmacy, Middle East University, Amman 11831, Jordan; mjsaadh@yahoo.com; 7Faculty of Pharmacy, Mutah University, Al-Karak 61710, Jordan

**Keywords:** empagliflozin, bioequivalence, pharmacokinetics, HPLC-MS/MS, adverse effect, tablet

## Abstract

The current study is a randomized, open-label, two-period, two-sequence, two-way crossover pharmacokinetic study in healthy Jordanian subjects to evaluate the pharmacokinetics and bioequivalence profile of two cases of empagliflozin 10 mg under fasting and fed conditions. The plasma concentrations of empagliflozin were determined using an HPLC-MS/MS method. Tolerability and safety were assessed throughout the study. This study included 26 subjects, 26 in both fasting and fed groups.The pharmacokinetic parameters, which included the area under the concentration–time curve from time zero to infinity (AUC_0–inf_) and the final quantifiable concentration (AUC_0–last_), maximum serum concentration (C_max_), and time to reach the maximum drug concentration (T_max_) were found to be within an equivalence margin of 80.00–125.00%. The pharmacokinetic profiles show that the empagliflozin test and parent reference cases were bioequivalent in healthy subjects. The two treatments’ safety evaluations were also comparable.

## 1. Introduction

Empagliflozin (Figure 1) is a competitive inhibitor of sodium–glucose co-transporter 2 (SGLT2) that is orally active and has an antihyperglycemic effect [1,2]. SGLT2 can be found in several anatomical areas in the body, including the proximal renal tubules in the kidneys, and it is capable of a more significant portion of reabsorption of excreted glucose out of the tubular lumen [3]. However, SGLT2 inhibitors can decrease glucose reabsorption and depress the renal threshold, elevating glucose in urine discharge [4]. Furthermore, it can be utilized to manage diabetes mellitus Type 2 (T2DM), besides food, and enhance glycemic control in individuals with type 2 diabetes [5]. Further, the use decreases cardiovascular disease hazard in a patient with T2DM and cardiovascular complications. For these purposes, it can be used as 10 mg orally every day and could be elevated to 25 mg/day if needed and afforded [6,7].

In the case of liver disability, there is no need to adjust the dose. However. in the case of kidney impairment, the following are advised: if eGFR (estimated glomerular filtration rate) ≥ 45 mL/min/1.73 m^2^, there is no necessary for dosage modification; however, if eGFR 30–45 mL/min/1.73 m^2^, there is a problem with starting therapy with this medication, but in the case that the therapy is already administered, then terminate therapy when eGFR is continuously <45 mL/min/1.73 m^2^. However, if eGFR < 30 mL/min/1.73 m^2^, treatment with this medication should be stopped [8,9].

Empagliflozin use has some contraindications. For example, it should be avoided in type 1 diabetes or diabetic ketoacidosis, where we have to evaluate kidney function before starting the medication and a specific time afterward, and in patients with volume depletion [10,11]. Several clinical studies have demonstrated empagliflozin’s efficacy and safety [12]. Empagliflozin was rapidly absorbed after single and multiple oral doses (0.5–800 mg), reaching peak plasma concentrations after approximately 1.33–3.0 h before exhibiting a biphasic decline. In single rising-dose studies, the mean terminal half-life ranged from 5.6 to 13.1 h, and in multiple-dose studies, it ranged from 10.3 to 18.8 h. Increases in exposure were dose-proportional after multiple oral doses, and trough concentrations remained constant after day 6, indicating that a steady-state had been reached. Oral clearance at a steady-state was comparable to single-dose values, indicating time-dependent linear pharmacokinetics. There were no clinically significant changes in pharmacokinetics in mild to a severe hepatic impairment or mild to severe renal impairment and end-stage renal disease. Clinical studies revealed no relevant drug–drug interactions with several drugs commonly prescribed to T2DM patients, including warfarin [11].

The peak plasma time was ninety minutes, and the plasma concentration will peak after 259 nmol/L (10 mg/day); 687 nmol/L (25 mg/day) and AUC: 1870 nmol hr/L (10 mg/day); 4740 nmol hr/L (25 mg/day) [1]. Empagliflozin could be used only via the oral route, either after or before food, and it is highly protein-bound (86.2%), limiting its distribution [13]. It has partitioning to RBCs of about 36.8%. The majority is metabolized through glucuronidation by the uridine 5′-diphospho-glucuronosyltransferases (UGT2B7) by the medication’s metabolism routes UGT1A3, UGT1A8, and UGT1A9 [14]. However, it has T_1/2_ elimination of about 12.4 h and clears at a rate of 10.6 L/hr. Empagliflozin is divided between urine and feces, at 54.4 and 41.2, respectively. Patients are asked to increase the drinking of fluids to decline the possibility of hypotension [15]. The best storage temperature for the drug is 25 °C (77 °F), with fluctuations permitted between 15–30 °C (59–86°F) [15].

The bioequivalence studies showed that at least several comparable dosage forms could cause absorption to the bloodstream with the same relative rate and extent [1]. Bioequivalence is usually used to assess a previously known two bioequivalence ranges. Moreover, bioequivalence studies use statistical tests of the hypothesis that evaluate the geometrical ratio and require similar times to achieve peak blood concentrations [16]. Empagliflozin does not have a primary metabolite, except for the most available glucuronide conjugates (2-O-, 3-O-, and 6-O-glucuronide) [17]. There is no comparative study of the bioequivalence of ten mg of empagliflozin of test and parent reference tablets in Jordanian fasting and fed conditions. Besides, the study aims to compare blood glucose reduction after taking empagliflozin in two cases: the original (Jardiance^®^, Boehringer Ingelheim, Germany) and generic forms. Furthermore, this study is of value for introducing new generic drugs in Jordan.

## 2. Results

### 2.1. Empagliflozin Analysis

Empagliflozin plasma contents were measured using a validated HPLC-MS/MS technique. As shown in Figure 2, the retention times for test and parent reference were 2.28 and 2.26, respectively.

### 2.2. Tolerability and Safety

Out of twenty-six subjects who participated, fifteen participants experienced AEs to study the fasting condition. Side effects were hypoglycemia, palpitations, fatigue, diarrhea, nausea, elevated bilirubin, hypotension, dizziness, hypertension, and sinus bradycardia. Three subjects developed a decrease in blood glucose levels, one of which was determined to be hypoglycemic, and after taking measures to give 20% glucose solution, hypoglycemia disappeared. In 16 subjects, there were 22 negative events in the fed condition and no adverse events related to hypoglycemia. Most treatment-emergent adverse events (TEAEs) were of the grade 1 variety, and all settled unexpectedly before the examination. The researchers considered all AEs to be mild or moderate because they were transient. There were no severe adverse events, such as death, reported. The AEs were associated with lowering the blood glucose of the two cases. No clinically substantial variations from the routine physical examination were noticed, including vital sign estimations and ECG recordings. Figure 3 depicts the findings of tracking the time plasma concentration profile of empagliflozin under fasting and fed conditions, as shown in Figure 3.

### 2.3. Pharmacokinetic Analysis

The pharmacokinetic study examined the mean empagliflozin drug level curves after dispensing a single dose of 10 mg tablets of two cases, the parent reference, and test, to 26 healthy Jordanian volunteers. Table 1 illustrates the basic pharmacokinetic properties of both drugs. The average standard deviation for C_max_ values of parent reference empagliflozin following oral treatment by the test preparations was 75.067 (26.315) g/L, and for parent reference preparations was 77.815 (21.131) g/L. Empagliflozin T_max_ value was 4.771 (1.610) h for the test and 4.75 (1.619) h for the parent reference, respectively. The extents of absorption, AUC_0–__ƚ__,_ and AUC_0–∞_, were 516.807 (149.78) and 539.198 (141.16) g/L/h after administration of the test case, and 596.551 (154.291) and 527.057 (139.525) g/L/h after administration of the parent reference drugs, respectively. In addition, t_1/2_ and V_d_ were 4.49 (1.36) h, 22.87 (8.99) L for the test preparations, and 4.00 (1.12), 19.07 (5.17) L for the parent reference cases.

Upon ANOVA analysis and using in-transformed data for empagliflozin, no sequence, period, or formulation effects were observed for any pharmacokinetic property (*p* > 0.05). Table 2 displays the 90 percent CIs of the proportions (test vs. parent reference) for the in-transformed C_max_, AUC_0–t_, and AUC_0–inf_. For empagliflozin, the 90% CIs for C_max_ ratios, AUC_0–t_, and AUC_0–inf_ were 95.69% to 116.99%, 92.77% to 110.01%, and 93.19% to 106.09%, respectively. These results meet the predefined bioequivalence requirements. The relative bioavailability of the test/parent reference preparation were 108.12% (mean C_max_), 98.40% (average AUC_0–t_), and 96.9% (average AUC_0–inf_), Table 2.

## 3. Discussion

The pharmacokinetics and bioequivalence of two cases of empagliflozin tablets were studied in healthy Jordanian volunteers of both sexes under fasting and fed conditions. The 90% CIs for empagliflozin were contained in previously determined bioequivalence standards of 80% to 125% for AUC and C_max_ [2]. Pharmacokinetics parameters and bioequivalence examination of the standard tablet in the two cases indicated that single-dose exposure following the two treatments was equivalent to C_max_, AUC_0–t_, and AUC_0–inf_ under fasting and fed conditions. Moreover, ANOVA study on the log-scale results T_max_, C_max_, and AUC_0–t_ and the untransformed data for C_max_, AUC_0–t_, AUC_0–inf_, t_1/2_, and T_max_ indicated that the sequence effects and cases or former effects for all these parameters did not affect the outcome of the study. No statistically significant differences were detected between the two cases (*p* > 0.05).

According to previous studies [18,19], giving empagliflozin with food resulted in mean C_max_ and AUC being depressed by 8% and 9%, respectively. Studies with single oral doses of empagliflozin in healthy individuals and multiple oral doses in patients with type 2 diabetes resulted in maximum drug concentration (C_max_) reached 2 to 3 h after administering the dose. However, this study showed that the mean C_max_ and AUC increased by approximately 50% to 74% when empagliflozin was given high-fat, high-calorie meals. Various factors may lead to the above differences, including differences between people, ethnicity, environment, and food intake pattern. At clinically relevant intestinal and systemic concentrations, empagliflozin does not inhibit CYPs 1A2, 2B6, 2C8, 2C19, 2D6, or 3A4. In vivo, empagliflozin is not expected to be a CYP2C9 inhibitor. Furthermore, empagliflozin and its glucuronides are not thought to be irreversible CYP inhibitors. As a result, drug–drug interactions involving the investigated CYPs are regarded as unlikely. Empagliflozin does not inhibit UGT1A1 at maximum organ concentrations [20]. However, it is challenging to deduce certain affecting factors based on the current study accurately.

Researchers used various drug delivery strategies versus FDA guidelines [21] and previous studies [21,22] in glucose administration for the fasting condition study. According to FDA rules, subjects were given a test or parent reference formula in 240 mL of 20 percent glucose solution and 60 mL of 20 percent glucose solution every 15 min for 4 h to reduce the risk of hypoglycemia while fasting empagliflozin bioequivalence analysis. Conversely, clinical research feedback indicates that Jordanian subjects are intolerant to a 20% glucose solution due to nausea and vomiting, interfering with the drug’s side effects. In this study, we gave empagliflozin 240 mL of 20% glucose solution and then 60 mL of 10% glucose solution every 15 min for the next four hours while closely monitoring hypoglycemia with blood glucose readings. When the subject develops hypoglycemia symptoms, 60 mL of 20% glucose solution is given.

Comparing the parent reference and test drug’s blood glucose concentration monitoring results under fasting or fed conditions may correlate with drug absorption. For the study of fasting conditions, blood glucose level reached the lowest near T_max_. In a healthy human body, the average blood glucose level of the parent reference and the test drugs was similar, reflecting the pharmacodynamic similarity of the parent reference and test drugs to a certain extent.

In a single-dose, open-label, randomized, crossover study of empagliflozin 10 mg in healthy male subjects, it was reported that drug-related incidence of hypoglycemia of 4.35% (1/23) [23]. In our fasting group of this study, drug-related hypoglycemia incidence was 2.78% (1/36). For empagliflozin studies that did not take 20% glucose solution early, the tolerance results differed from current studies [24,25]. A single-dose, open-label, randomized, two-sequence, two-period crossover pharmacokinetic study with 10 mg empagliflozin reported that drug-related incidence of hypoglycemia was 25.0% (6/24) [23].

In another randomized, double-blind, placebo-controlled monotherapy study with a fourteen-week trial, patients already on sulfonylurea therapy undergoing a three-week washout interval were randomized to the parent reference of 10 mg, 25 mg, or placebo. Patients randomized to a reference of 10 or 25 mg undergoing forced titration from an initial dose of 10 mg to final doses, as tolerated. The overall occurrence of any hypoglycemia was 4% for empagliflozin 25 mg, 17% for empagliflozin 10 mg, and 0% for placebo [26]. According to the drug description [18], empagliflozin may cause adverse reactions to hypoglycemia when administered under fasting conditions. Therefore, it is necessary to monitor the blood glucose concentration in studying the clinical bioequivalence of drugs related to diabetes treatment.

The current study had a sufficient number of subjects to ensure adequate statistical power to prove the equivalence of the test product to the parent reference product. However, the study has some limitations. Due to limited recruitment capacity, we could only include male volunteers. Empagliflozin is not recommended in women trying to conceive because Empagliflozin is rated a safety category C in pregnancy [1], and no women who cannot conceive volunteered to participate in the study.

## 4. Materials and Methods

### 4.1. Participants

Twenty-six individuals participated in this research to study under fasting conditions. One subject was excluded due to unqualified vital signs in the second period. Conversely, 26 subjects were enrolled under fed conditions. Moreover, one subject was excluded from the study due to taking nonexperimental drugs because of toothache in the second period. Dropped subjects did not take the second-period trial drug in both two conditions. Table 3 shows the demographics of the participants in the research.

### 4.2. Study Design

Bioequivalence testing was performed using an open-label single dose, random sequence, two × two crossover approach. Through doses, a one-week washout interval assesses the bioequivalence of two brand names, empagliflozin, manufactured as 10 mg tablets by the producer Boehringer Ingelheim (Jardiance^®^, Ingelheim, Germany; Batch number, 1705039) as a reference, with the other brand drug Test (Amman, Jordan; Batch number, B615).

Approval Number EC-UOP/101-12020 was obtained by adhering to the Declaration of Helsinki’s ethical standards for human research and the International Conference on Harmonization’s Good Clinical Practice Guideline and the NMPA’s Guideline for Good Clinical Practice [27].

### 4.3. Subjects

This study participants were healthy Jordanian volunteers at some private hospitals (Amman, Jordan) Phase I Clinical Unit. Before starting the study, complete medical and laboratory tests were obtained to ensure health. Smokers, heavy drinkers, those who used CYP enzyme inhibitors within the previous 60 days, those who had taken any medicine within the previous four weeks, those who had a history of medication allergies, those who had participated in previous clinical studies within the previous six months, and those with any significant clinical abnormality were all ruled out.

For the study of the fasting condition, volunteer age was between 23 and 49 years, they had a mean body mass of 78.45 kg, and body mass index ranged from 18.15 to 27.60 kg/m^2^. For the fed condition study, volunteer age was between 19 and 43 years, their mean body mass was 63.26 kg, and their body mass index ranged from 18.56 to 26.71 kg/m^2^. All subjects were knowledgeable regarding details, including the current study’s threats and advantages, and informed documentary agreements before starting the study. Subjects had the option to drop out of the study at any time.

### 4.4. Empagliflozin Dosing

This two-sequence crossover study compares the pharmacokinetics of two oral drugs of empagliflozin at a dose of 10 mg in healthy sex. A random number table was generated using SAS statistical software (version 9.130, SAS, Cary, NC, USA), and subjects were divided into T/R or R/T groups. A test or reference drug tablet containing 240 mL of 20% glucose solution was administered in a standing posture under fasting conditions. Under fed conditions, a high-fat (about 50%), high-calorie (800~1000 kcal) meal was consumed 30 min before the drug was administered, and the drugs were administered in 240 mL warm water.

### 4.5. Blood Sampling Plan

Venous blood samples (four mL per sample) were obtained before giving the medication (hour 0) and at 0.5, 1, 1.5, 2, 2.5, 3, 3.5, 4, 4.5, 5, 6, 7, 8, 10, 12, 14, 16, 24, 48 and 72 h. Quickly, samples were centrifuged at 3500× *g* rpm for seven min, and serum was extracted and frozen at −65 ± 20 °C within 120 min. Analysis of blood glucose content was followed regularly at 60 ± 15, 120 ± 30, 150 ± 30, 180 ± 30, and 240 ± 30 min after empagliflozin ingestion both in the conditions of fasting and fed [28].

### 4.6. Assay Method

The plasma concentrations of empagliflozin were determined using a revised version of the proven HPLC-MS/MS method [29]. The main instruments used in this study were an API—1400 mass spectrometer with a built-in waste/detector switching valve (AB-Sciex, CA, USA) and an HPLC system (Agilent Technologies, model LC-1200, Englewood, Colorado, USA) with an auto-sampler and controlled by Analyst 1.6.1 software. Bath sonicated Crest model-175T (UltraSonics CORP, Trenton, NJ, USA), Sartorius balance BP 2215, Eppendorf Centrifuge, and Windows XP SP3 Data Management Software 1.5.2-a were used. Multiple reaction monitoring transitions were observed at mass-to-charge ratios (*m*/*z*) of 461.3 → 449.2 and 465.6 → 440.9 for empagliflozin and 467.4 → 432.7 for d5-empagliflozin. Data acquisition and processing were powered by the Analyst 1.6.3 software package and Watson LIMS 7.5spl. Using a mobile phase of 70% methanol and 30% a mixture of 20 mM ammonium acetate and 0.2 mM formic acid, the chromatographic conditions of LC-MS/MS were quantitatively improved for the best analytical peak quality and shortest run time. They were isocratically eluted at 1 mL min-1 through an ACETM C18 (50 2.1 mm, 5 m) column for LC-MS/MS. Both injection volumes were two microliters in size. For empagliflozin analysis in ESI negative mode, the following MS parameters were optimized: nitrogen gas one flow = 60 units, gas two flow = 75 units, curtain gas = 35 units, ion spray voltage = 5000 Volts, drying temperature = 650 °C, and collision energy = 20 Volts [30].

Quality control samples were prepared at concentrations greater than five times the upper limit of quantification (5 × uloq) and then diluted five times with blank plasma. Six samples were prepared parallel with diluted sample concentrations within the standard curve’s linear range. The mean SD of the recovery rate of six parallel samples of the same concentration describes dilution accuracy. The average deviation of each concentration sample’s accuracy ranges between 85.00 and 115.00 percent, with a CV percent of 15.00 percent. The stability test items included the stability of drug-containing plasma after 24 h at room temperature and 24 h of treatment in an environment ranging from 2 to 8 degrees Celsius. Furthermore, four freeze–thaw cycles of drug-containing plasma at −30 to −10 °C and −80 to −60 °C, and 66 days of long-term storage of drug-containing plasma from −30 to −10 °C and −80 to −60 °C were performed. In the conditions mentioned above, the stability of empagliflozin was acceptable. Empagliflozin had a linear range of 0.5000 to 150.0 ng/mL. The standard curve LLOQ was from −7.98% to 10.46%, the accuracy deviation of the other concentration samples except LLOQ was from −12.00% to 12.88%, and R^2^ was from 0.9932 to 0.9995.

### 4.7. Pharmacokinetics and Statistical Analysis

The pharmacokinetics study used a non-compartmental method using Phoenix Win-01-Nonlin version 7.01 (Pharsight^®^, Princeton, NJ, USA). Blood concentration–time data were collected after fasting or fed administration. The pharmacokinetics (AUC, C_max_, T_max_) were statistically analyzed, and bioequivalence was evaluated. The AUC_0−t_ and AUC_0–inf_ for empagliflozin dugs were calculated by the trapezoidal method. The T_max_ values of the test (T) and reference drugs (R) were analyzed by the nonparametric Wilcoxon method. The point estimates of T_max_, C_max_, and AUC of test and reference drugs R were calculated after logarithmic conversion, and the significance test of T_max_, C_max_, and AUC was carried out by single-factor analysis of variance (ANOVA). Then, the statistical treatment of double unilateral *t* test was carried out. If the 90% confidence interval values of AUC_0−t_, AUC_0–inf_, and C_max_ geometric mean ratio were between 80.00% and 125.00% of the statistical interval proposed by the NMPA, the test drug was bioequivalent to the reference one.

### 4.8. Tolerability and Safety

Throughout the study, safety was evaluated using adverse events (AEs) and laboratory tests (biochemistry, hematology, and urinalysis). Potential adverse events (AEs) and vital signs (systolic and diastolic blood pressure, body temperature, and pulse rate) were assessed. Meanwhile, researchers assessed and recorded AEs in terms of seriousness, intensity, time course, outcome, and relationship to the study drug throughout the study. AEs have designated codes with a preferred term and system organ class, according to the Medical Dictionary for Regulatory Activities (version 20.0). Five description types (unrelated, unlikely, possibly, probably, or related) were recorded to confirm the relationships between AEs and drugs. 

## 5. Conclusions

The current pharmacokinetics study found that the two drugs were bioequivalent when fasted and fed. According to the guidelines, the central pharmacokinetics were within the bioequivalence range (80.0 to 125.0 percent). Two empagliflozin cases were well tolerated. The bioequivalent form of the 10 mg oral tablet will provide Jordanian patients with affordable, acceptable, and beneficial access to their medication. However, the pharmacokinetic changes under fasting and fed conditions differed from previous studies. This study may provide a new way to conduct better clinical pharmacokinetics and bioequivalence research of drugs related to diabetes treatment under fasting conditions, in addition to taking a lower glucose concentration to reduce intolerance. As a take-home message, oral administration of Empagliflozin under fasting and fed conditions resulted in equivalent pharmacokinetics, and thus both products could be considered bioequivalent. Because of their predictable bioavailability and low toxicity, both drugs could be considered interchangeable for patients seeking antihyperglycemic relief.

## Figures and Tables

**Figure 1 pharmaceuticals-15-00193-f001:**
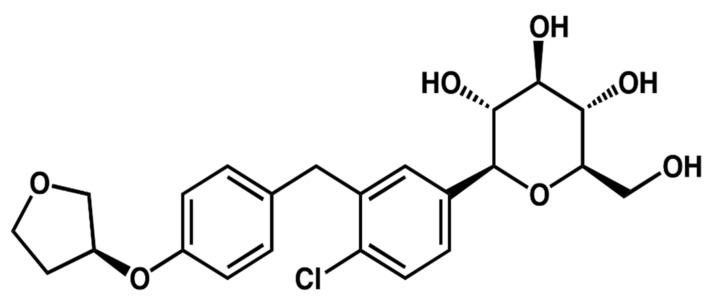
Chemical structure of empagliflozin.

**Figure 2 pharmaceuticals-15-00193-f002:**
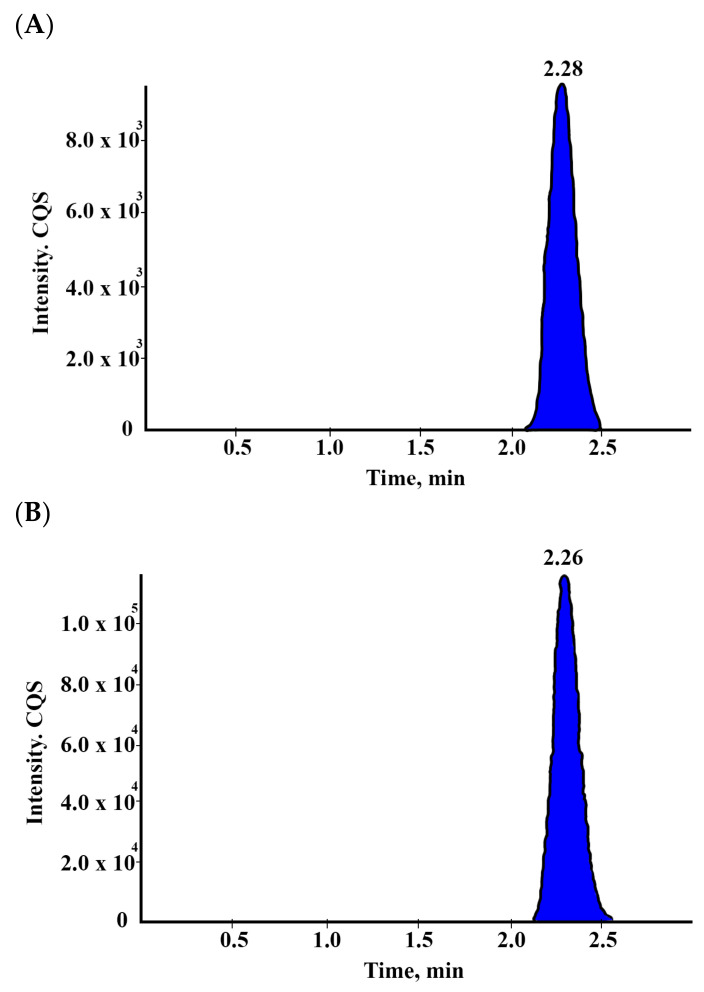
Chromatograms of the empagliflozin test (**A**) and parent reference (**B**) were 2.28 and 2.26, respectively.

**Figure 3 pharmaceuticals-15-00193-f003:**
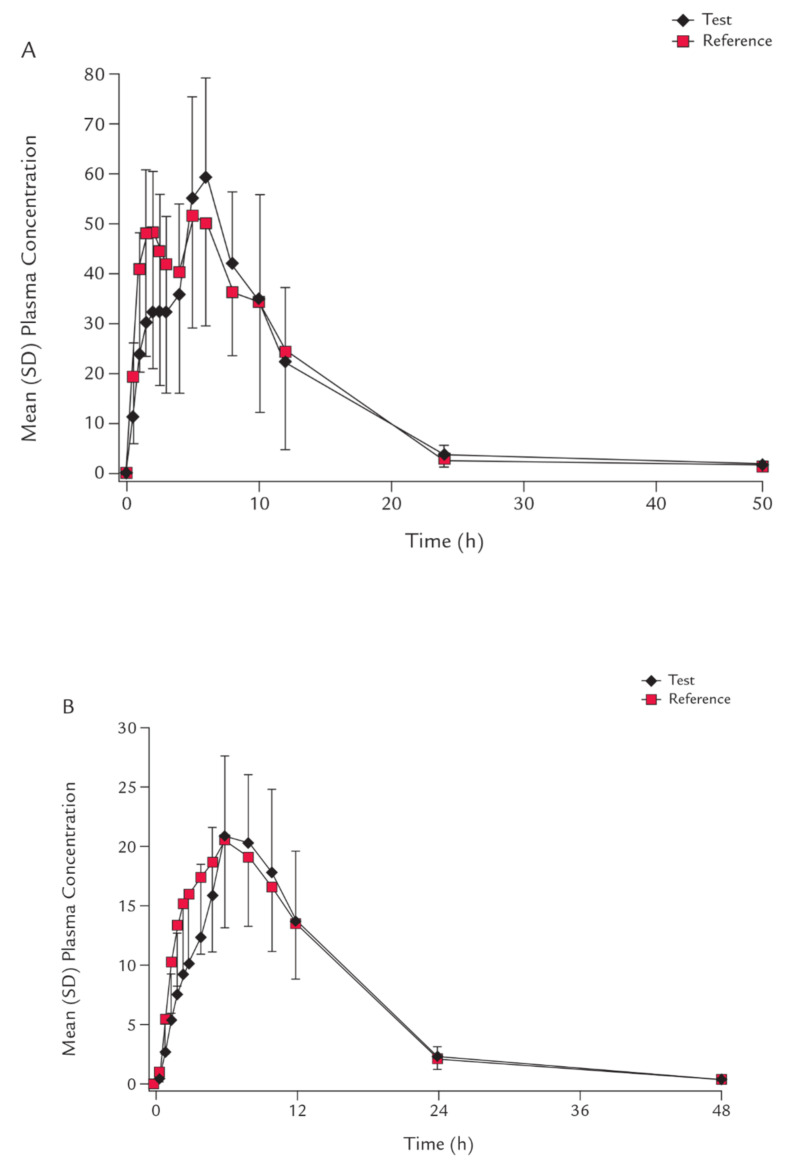
The time plasma concentration profile of empagliflozin under fasting (**A**) and fed (**B**) condition states in nmol/L.

**Table 1 pharmaceuticals-15-00193-t001:** Empagliflozin pharmacokinetic parameters after a single 10 mg in two cases; male volunteers (*n* = 26). Data are mean (SD).

Parameters	Parent Reference Empagliflozin	
	Test	Parent Reference
C_max_ μg/L	75.06 (26.31)	77.81 (21.23)
T_max_ (h)	4.77 (1.61)	4.75 (1.619)
AUC_0–__ƚ_ (μg/L/h)	516.80 (149.78)	596.55 (154.29)
AUC_0–∞_ μg/L/h	539.19 (141.16)	527.05 (139.52)
t_1/2_ (h)	4.49 (1.36)	4.00 (1.12)
V_d_ (L)	22.87 (8.99)	19.07 (5.17)

**Table 2 pharmaceuticals-15-00193-t002:** Correlation of 90 percent CIs of Ln-transformed variables for the parent reference empagliflozin after administration of two empagliflozin tablet formulations (test/parent reference).

Compound/Parameter	Test/Parent Reference (%)	90% CI	*p*	
			<80%	>125%
Parent Reference Empagliflozin				
In C_max_	108.12	95.69–116.99	0.001	0.0029
In AUC_0–__ƚ_	98.40	92.77–110.01	<0.001	<0.001
In AUC_0–∞_	96.9	93.19–106.09	<0.001	<0.001

**Table 3 pharmaceuticals-15-00193-t003:** Characteristics of the study’s participants.

Characteristic	Mean ± SD	Range
Age (Year)	34.9 ± 1.7	22.7–57.3
Weight (Kg)	79.5 ± 9.8	66.7–92.4
Height (cm)	174.8 ± 8.8	161.7–189.1

## Data Availability

The data presented in this study are available in article.

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
