# Peer review of "Pharmacokinetics and Bioequivalence of Two Empagliflozin, with Evaluation in Healthy Jordanian Subjects under Fasting and Fed Conditions"

_pharmaceuticals, 2022, doi:10.3390/ph15020193_

Round 1
Reviewer 1 Report
The Hailat et al. research was a randomized, open-label, two-period, two-sequence, two-way crossover pharmacokinetic study in healthy Jordanian subjects to evaluate the pharmacokinetics and bioequivalence profile of two formulations of empagliflozin 10-mg under fasting and fed conditions.
The study covers some issues that have been overlooked in other similar topics. The structure of the manuscript appears adequate and well divided in the sections. Moreover, the study is easy to follow, but some issues should be improved. The manuscript needs moderate grammar correction. Please also check typos thorough the text.
Overall manuscript including: This manuscript required a general revision to 1-) eliminate redundant sentences and to 2-) add limitation of the study in the discussion section, as well 3-) some "take-home message" in the conclusion section.
Author Response
The Hailat et al. research was a randomized, open-label, two-period, two-sequence, two-way crossover pharmacokinetic study in healthy Jordanian subjects to evaluate the pharmacokinetics and bioequivalence profile of two formulations of empagliflozin 10-mg under fasting and fed conditions.
The study covers some issues that have been overlooked in other similar topics. The structure of the manuscript appears adequate and well divided in the sections. Moreover, the study is easy to follow, but some issues should be improved. The manuscript needs moderate grammar correction. Please also check typos thorough the text.
Overall manuscript including: This manuscript required a general revision to 1-) eliminate redundant sentences and to 2-) add limitation of the study in the discussion section, as well 3-) some "take-home message" in the conclusion section.
We are indebted to you for taking the time to review and improve our manuscript. Your astute observations are greatly appreciated. The manuscript was significantly improved due to reviewer comments, suggestions, and feedback.
The reviewer's comment is adopted, and a native speaker reads and evaluates the writing's quality. We appreciate your suggestion.
In addition, we did eliminate redundant sentences for our best-discovered sentences and added a limitation sentence of the study after the discussion section, as well as we added some "take-home messages" after the conclusion section.
Finally, we would like to express our gratitude and appreciation for the reviewer's effort and time spent improving and cleaning the manuscript—many thanks.
Reviewer 2 Report
The paper entitled "Pharmacokinetics and Bioequivalence of two Empagliflozin, with Evaluation in Healthy Jordanian Subjects under Fasting and Fed Conditions" is well written with methods thoroughly described. Results are appropriately presented and discussed. Conclusions are supported by the results. A small number of typing errors was detected - eg. numbering of headings. The matters of concern might be the significance of content and interest to readers.
Author Response
The paper entitled "Pharmacokinetics and Bioequivalence of two Empagliflozin, with Evaluation in Healthy Jordanian Subjects under Fasting and Fed Conditions" is well written with methods thoroughly described. Results are appropriately presented and discussed. Conclusions are supported by the results. A small number of typing errors was detected - eg. numbering of headings. The matters of concern might be the significance of content and interest to readers.
We appreciate the reviewer's time and effort in thoroughly reading and reviewing our article. The reviewer enquired about "the significance of content and reader interest." While it refers to Jordanians who were available to conduct our study, it can be extended to other populations with comparable cases, as documented in the literature (al Bawab et al., 2020; W. Li et al., 2021; Y. Li et al., 2020; Shao et al., 2021). Additionally, this study can be used as a benchmark for their global research. We appreciate the reviewer's input.
al Bawab, A. Q., Alkhalidi, B. A., Albarahmieh, E., Qassim, S. M. A., Al-Saifi, M. A. D., Al-Saifi, B., Ling, J., & Al-Qerem, W. (2020). Pharmacokinetics and Bioequivalence Estimation of Two Formulations of Alfuzosin Extended-Release Tablets. Clinical Pharmacology in Drug Development, 9(7), 780–784. https://doi.org/10.1002/CPDD.860
Li, W., Wang, Y., Pei, Y., & Xia, Y. (2021). Pharmacokinetics and Bioequivalence Evaluation of Two Montelukast Sodium Chewable Tablets in Healthy Chinese Volunteers Under Fasted and Fed Conditions. Drug Design, Development and Therapy, 15, 1091–1099. https://doi.org/10.2147/DDDT.S298355
Li, Y., Qi, L., Bai, H., Liu, Y., Fan, R., Tu, Y., Sun, Y., Wang, J., Qi, Q., Feng, X., Zhou, D., & Wang, X. (2020). Pharmacokinetics and Bioequivalence of Rasagiline Tablets in Chinese Healthy Subjects Under Fasting and Fed Conditions: An Open, Randomized, Single-Dose, Double-Cycle, Two-Sequence, Crossover Trial. Frontiers in Pharmacology, 11, 1938. https://doi.org/10.3389/FPHAR.2020.571747/BIBTEX
Shao, R., Yang, D. dan, Ruan, Z. rong, Chen, J. liang, Hu, Y., Jiang, B., & Lou, H. gang. (2021). Pharmacokinetic and Bioequivalence Evaluation of 2 Tadalafil Tablets in Healthy Male Chinese Subjects Under Fasting and Fed Conditions. Clinical Pharmacology in Drug Development. https://doi.org/10.1002/CPDD.1007
Reviewer 3 Report
The authors presented a bioequivalence study of empagliflozin in fed and fasted conditions. The study design is well designed but the results are poorly presented and need substantial improvement.
- The two formulations (test and reference) of empagliflozin were administered in fed and fasted conditions. The pharmacokinetic and statistical results of those two conditions should be presented separately.
- I do not think that figure 3 presents the blood glucose levels of subjects. It seems the time plasma concentration profile of empagliflozin under fasting and fed condition.
- I can not understand why the authors presented table 3. Because the same doses are given to all of the subjects it does not need dose normalization. Furthermore, the bioequivalence test compares pharmacokinetic parameters within the subjects, it does not need weight normalization to compare pharmacokinetic parameters.
Author Response
The authors presented a bioequivalence study of empagliflozin in fed and fasted conditions. The study design is well designed but the results are poorly presented and need substantial improvement.
- The two formulations (test and reference) of empagliflozin were administered in fed and fasted conditions. The pharmacokinetic and statistical results of those two conditions should be presented separately.
We concur with the reviewer, and the results of the test and reference formulations were presented separately for clarity. We appreciate the reviewer's suggestion.
- I do not think that figure 3 presents the blood glucose levels of subjects. It seems the time plasma concentration profile of empagliflozin under fasting and fed condition.
Correct. Figure 3 has been updated to reflect the corrected format. We appreciate the reviewer identifying and alerting us to this error.
- I can not understand why the authors presented table 3. Because the same doses are given to all of the subjects it does not need dose normalization. Furthermore, the bioequivalence test compares pharmacokinetic parameters within the subjects, it does not need weight normalization to compare pharmacokinetic parameters.
We appreciate your assiduous reading and noticing that. As a result, the entire Table was deleted. We appreciate you noticing and informing us of this.
We appreciate your insightful comments once again. We feel much more confident about the manuscript's contents, findings, and organization due to your feedback.
Finally, we would like to express our gratitude and appreciation for the reviewer's effort and time spent improving and cleaning the manuscript—many thanks.
Reviewer 4 Report
A few sentences like: „However, it has the T1/2elimination of about 63 12.4 hrs and cleared at a rate of 10.6 L/hr.” require corrections.
Fig. 2 – its resolution should be improved.
Authors should explain the impact of the CYP2C9 genotype on empagliflozin metabolism if they mentioned it.
In general, there is too much data on the use of the drug by patients and much less about its ADME or ADME-Tox properties.
The term formulation was used all over the manuscript but in some fragments, I got the impression that in fact, the difference was in dose (10 and 25mg), not in the formulation. Correct this.
What are these two formulations in detail? Differences between these two forms of the drug are only mentioned in ‘Study design’ but should be described in more detail. Misleadingly, only one is described in the part entitled: “Empagliflozin l Analysis”. It seems that these two formulations are the original and generic form of Empagliflozin but it needs to be described in Introduction not in Methods.
Also, it is rather difficult to spot which data refer to which form of the drug. Authors use term ‘Parent Empagliflozin’ and ‘test’ or ‘reference’. It should be corrected all over the manuscript, e.g., to Empagliflozin form 1 and Empagliflozin form 2.
Discussion and Conclusion part should be extended because it is not clear if differences between these two formulations are indeed crucial for patients.
There is also nothing about the actual differences between these two formulations. What are their compositions, are there any modifications of APIs in them? Are there any additional compounds that could have impact on the drug pharmacokinetics properties?
Author Response
We are indebted to you for taking the time to review and improve our manuscript. Your astute observations are greatly appreciated. The manuscript was significantly improved due to reviewer comments, suggestions, and feedback.
The reviewer's comment is adopted, and a native speaker reads and evaluates the writing's quality. We appreciate your suggestion.
A few sentences like: „However, it has the T1/2elimination of about 63 12.4 hrs and cleared at a rate of 10.6 L/hr.” require corrections.
The reviewer's comment is adopted, and a native speaker reads and evaluates the writing's quality. We appreciate your suggestion.
Fig. 2 – its resolution should be improved.
The reviewer's comment was adopted, and the resolution of the figure was increased. Thank you for this comment.
Authors should explain the impact of the CYP2C9 genotype on empagliflozin metabolism if they mentioned it.
We sincerely appreciate and thank you for your insightful comment and concern. This comment informs us of a serious error in the article: "Another factor contributing to the differences could be the CYP2C9 genotype [21], affecting empagliflozin metabolism." First and foremost, please accept our apologies for this blunder. Second, we updated the statement to reflect the correct information. Thank you for bringing this to our attention.
In general, there is too much data on the use of the drug by patients and much less about its ADME or ADME-Tox properties.
Thank you for bringing this up. A sentence about the dug's ADME or ADME-Tox properties was added accordingly. Thank you for your input.
The term formulation was used all over the manuscript but in some fragments, I got the impression that in fact, the difference was in dose (10 and 25mg), not in the formulation. Correct this.
Thank you for informing us of this. To reflect the correct meaning, the word "formulation" was replaced with the appropriate word, such as drug, case, etc. Thank you for your comment.
What are these two formulations in detail? Differences between these two forms of the drug are only mentioned in ‘Study design’ but should be described in more detail. Misleadingly, only one is described in the part entitled: “Empagliflozin l Analysis”. It seems that these two formulations are the original and generic form of Empagliflozin but it needs to be described in Introduction not in Methods.
This is correct, and the reviewer's suggestion was adopted, with a sentence added to the introduction to clarify the word "formulation." Thank you for your suggestions.
Also, it is rather difficult to spot which data refer to which form of the drug. Authors use term ‘Parent Empagliflozin’ and ‘test’ or ‘reference’. It should be corrected all over the manuscript, e.g., to Empagliflozin form 1 and Empagliflozin form 2.
The ambiguity was removed in this regard by including the name Jardiance® (Boehringer Ingelheim, Germany) in the introduction, as suggested by the reviewer. The feedback is greatly appreciated and thanked.
Discussion and Conclusion part should be extended because it is not clear if differences between these two formulations are indeed crucial for patients.
We appreciate your comment. Both formulations are extremely similar, but one is the original brand, and the other is a copy of it. We appreciate your comment.
There is also nothing about the actual differences between these two formulations. What are their compositions, are there any modifications of APIs in them? Are there any additional compounds that could have impact on the drug pharmacokinetics properties?
We appreciate your input. Both formulations are extremely similar, but one is the original brand and the other is a copy of it. We appreciate your input.
We appreciate your insightful comments once again. We feel much more confident about the manuscript's contents, findings, and organization due to your feedback.
Finally, we would like to express our gratitude and appreciation for the reviewer's effort and time spent improving and cleaning the manuscript—many thanks.
Round 2
Reviewer 3 Report
Thanks for revising the manuscript. I think it is now appropriate for the publication.